Using citizen-science data to identify local hotspots of seabird occurrence

Ward Eric J. 1 eric.ward@noaa.gov
Marshall Kristin N. 1
Ross Toby 2 3
Sedgley Adam 2 3
Hass Todd 4 5
Pearson Scott F. 6
Joyce Gerald 3 7
Hamel Nathalie J. 3 8
Hodum Peter J. 3 9
Faucett Rob 3 5
1 Conservation Biology Division, Northwest Fisheries Science Center, National Marine Fisheries Service, National Oceanic and Atmospheric Administration , Seattle, WA , USA
2 Seattle Audubon Society , Seattle, WA , USA
3 Science Committee, Seattle Audubon Society , Seattle, WA , USA
4 School of Marine and Environmental Affairs , Seattle, WA , USA
5 Burke Museum of Natural History and Culture, University of Washington , Seattle, WA , USA
6 Wildlife Science Division, Washington Department of Fish and Wildlife , Olympia, WA , USA
7 Moon Joyce Resources , Seattle, WA , USA
8 Puget Sound Partnership , Tacoma, WA , USA
9 Biology Department, University of Puget Sound , Tacoma, WA , USA
Gandini Patricia
Electronic publication date: 2015 Jan 15
Publication date: 2015
Volume: 3
Electronic Location ID: e704
Received 2014 Oct 2; Accepted 2014 Nov 28
Copyright: © 2015 Ward et al.
Copyright year: 2015
Copyright holder: Ward et al.
License: This is an open access article distributed under the terms of the Creative Commons Attribution License, which permits unrestricted use, distribution, reproduction and adaptation in any medium and for any purpose provided that it is properly attributed. For attribution, the original author(s), title, publication source (PeerJ) and either DOI or URL of the article must be cited.
License URL: https://creativecommons.org/licenses/by/4.0/

Keywords: Puget Sound, Seabirds, Citizen-science, Hotspots, Spatial models, Occupancy models, Salish Sea

Funding: Boeing Sustainable Path Foundation Russell Family Foundation WDFW Patagonia Funding for the Puget Sound Seabird Survey was provided by Boeing, Sustainable Path Foundation, Russell Family Foundation, WDFW, and Patagonia. The funders had no role in study design, data collection and analysis, decision to publish, or preparation of the manuscript.

==============================
Seabirds have been identified and used as indicators of ecosystem processes such as climate change and human activity in nearshore ecosystems around the globe. Temporal and spatial trends have been documented at large spatial scales, but few studies have examined more localized patterns of spatiotemporal variation, by species or functional group. In this paper, we apply spatial occupancy models to assess the spatial patchiness and interannual trends of 18 seabird species in the Puget Sound region (Washington State, USA). Our dataset, the Puget Sound Seabird Survey of the Seattle Audubon Society, is unique in that it represents a seven-year study, collected with a focus on winter months (October–April). Despite historic declines of seabirds in the region over the last 50 years, results from our study are optimistic, suggesting increases in probabilities of occurrence for 14 of the 18 species included. We found support for declines in occurrence for white-winged scoters, brants, and 2 species of grebes. The decline of Western grebes in particular is troubling, but in agreement with other recent studies that have shown support for a range shift south in recent years, to the southern end of California Current.

Introduction

Ecologists and conservation practitioners have long focused on describing species distribution and estimating changes in abundance (Holmes, 2001) or occurrence through time (MacKenzie et al., 2006). Using species distribution modeling to identify the spatial variability and hotspots of a species’ distribution has multiple implications for science and management. From a conservation perspective, incorporating spatial variation in models may assist in selecting areas to protect or predicting where species are likely to persist (Cabeza & Moilanen, 2001; Naujokaitis-Lewis et al., 2009). From a theoretical ecology perspective, null or neutral models of species’ occurrence may be useful in predicting species diversity or community assembly (Gotelli, 2000; Gotelli & McGill, 2006). Finally, the inclusion of spatial variation has implications for management and policy in that accounting for spatial heterogeneity may help in forecasting how species may respond to future environmental conditions, such as range shifts in response to climate change (Jetz, Wilcove & Dobson, 2007).

In addition to quantifying spatial variation, species distribution modeling can be used to assess temporal trends in occurrence, which themselves may be spatially structured as well. Mechanisms responsible for spatially structured trends may include changing habitat conditions, behavior, or prey availability (Ward et al., 2010). Spatially structured anthropogenic disturbances (e.g., wildfires, oil spills, climate change, urbanization) may have similar impacts, and collectively ignoring such underlying spatial variation when it exists may lead to poor estimation of temporal trends (Hoeting et al., 2006).

Models that incorporate both spatial and temporal variation represent a rapidly evolving field in ecology (Hooten & Wikle, 2008; Latimer et al., 2006; Shelton et al., 2014). While many of these methods have been in the statistical literature for decades (Banerjee, Gelfand & Carlin, 2005; Cressie & Wikle, 2011), ecological data often present a unique set of challenges relative to data from other fields. Compared to other disciplines, ecological data on species abundance is often corrupted by observation error, which represents uncertainty arising from taking measurements or sampling a fraction of the population (Holmes, 2001). Similarly, in conducting studies of species presence–absence, detections may be missed, resulting in false-negatives (MacKenzie et al., 2006). Because of recent computational advances, statistical models that include both spatial and temporal variation are now widely available to ecologists and offer a powerful tool for assessing changes in species distributions through time.

As data hungry methods have advanced, monitoring programs have suffered in recent years because of declining budgets and an increased need for cost efficient survey techniques. In the face of recent reductions in monitoring programs, one potentially underutilized resource is citizen science, which may be a useful tool for conducting baseline environmental monitoring or helping to inform management actions or restoration activities (Cooper et al., 2007). Participation in these volunteer-based programs appears to have increased in recent years (Silvertown, 2009). Some of the longest running citizen-science programs in North America are related to bird watching. Large-scale volunteer programs like the Audubon Christmas Bird Count and the North American Breeding Bird Survey (BBS) have been effective at collecting vast amounts of survey data on commonly occurring bird species (Sauer et al., 2014). The strength of these programs is their duration, large spatial extent, and consistent methodologies over time, enabling them to be useful in monitoring species assemblages and distribution shifts in response to changing climate (Hitch & Leberg, 2007).

In the Pacific Northwest, similar regional-scale citizen-science programs focused on seabirds have been established. These include the Coastal Observation and Seabird Survey Team (COASST) (Hamel et al., 2009; Litle, Parrish & Dolliver, 2007; Parrish et al., 2007) and the British Columbia Coastal Waterbird Survey (Crewe et al., 2012), which have also been developed to address conservation questions and establish baseline monitoring. These regional citizen-science programs help inform restoration actions in Puget Sound (Washington State), where one of the largest ecosystem restoration programs in the nation is underway (Puget Sound Partnership, 2014). The Puget Sound ecosystem is part of the Salish Sea (which also includes the Strait of Juan de Fuca and the Strait of Georgia), and has been affected by widespread environmental degradation largely associated with increased urbanization (effects summarized in Puget Sound Partnership, 2009; Ruckelshaus & McClure, 2007). Puget Sound consists of over 4,000 km of coastline, with a suite of high-value ecosystem services, including commercial fisheries and various recreation opportunities (e.g., Tallis & Polasky, 2009). A portfolio of ecosystem indicators has been developed and incorporated into restoration goals for the Puget Sound region to monitor ecological conditions, including seabirds (Kershner et al., 2011; Puget Sound Partnership, 2013).

A limitation of using the data from many citizen-science programs—including those from regional seabird monitoring programs in the Pacific Northwest—has been that survey effort is generally not quantified. A limited number of agency-funded seabird surveys have been conducted that allow the assessment of trends. To assess winter seabird densities, the Washington Department of Fish and Wildlife (WDFW) has conducted annual aerial surveys since 1992, representing a 6-week snapshot of density (Anderson et al., 2009; Nysewander et al., 2005). These annual transects occur in 13–18% of the nearshore (<20 m depth) and 3–6% of the offshore (>20 m depth) marine waters in Puget Sound, ranging from southern Puget Sound to the Canadian border. Results from the WDFW aerial seabird surveys suggested that the density of some species, including Western Grebes (Aechmophorus occidentalis), has declined over the last two decades (Bower, 2009; Evenson, 2010; Vilchis et al., 2014). However, the cause(s) of these declines and the effects of environmental drivers on seabird density remain largely unknown. To complement the WDFW seabird survey both spatially and temporally, and to establish further baseline monitoring of local seabird species occurrence and abundance in winter months, the Seattle Audubon Society initiated the shore-based Puget Sound Seabird Survey (PSSS) in 2007. This program is unique in Puget Sound, in that volunteers monitor study sites in nearshore habitats monthly, from late fall to early spring (October–April). The October–April period was chosen because this is the window of greatest seabird abundance and diversity for this ecosystem. This survey also represents a good example of a scientifically rigorous citizen-science effort because survey effort is quantified and volunteers are trained annually and are the subject of ongoing validation studies to quantify biases, such as species misidentification.

Recent research has demonstrated that rigorous statistical models can be applied to volunteer-based surveys, yielding a relatively large impact, particularly when agency or industry-led data collection efforts are limited (Thorson et al., 2014). The primary objective of our analysis was to apply spatiotemporal models to data from the Puget Sound Seabird Survey to (1) evaluate relative hotspots of occurrence over the period 2007–2014, (2) evaluate temporal trends in occupancy over this period, and (3) establish a baseline for future monitoring in the region. These spatial and temporal estimates of occupancy may also be useful to refine the list of indicator species used to quantify ecosystem change or restoration progress.

Methods

PSS survey data collection

Beginning in October 2007, teams of 2–4 volunteer birdwatchers were trained by Seattle Audubon staff to collect data on birds in the nearshore environment of Puget Sound. Though the species encountered includes waterfowl, we collectively refer to all species as ‘seabirds.’ Each observer team was responsible for monthly surveys (October–April) at selected sites. Many of the seabird species in the region overwinter in Puget Sound, and are of highest abundance in late fall—early spring. The PSSS survey sites were selected non-randomly due to dependence on public access (parks, beach access), but they were selected to be spaced at least 1.6 km apart. Observer teams recorded all species present out to 300 m from shore for a minimum of 15 min, but some site visits lasted up to 60 min. To minimize the variability of weather conditions, tidal stage, and the risk of double counting birds at multiple survey sites, volunteer teams completed their monthly surveys on the same date within a specific four-hour window (two hours on either side of daylight high tide) on the first Saturday of each month. In each subsequent year of surveys, we added sites to cover parts of northern and southern Puget Sound. For this study, we limited our analysis to 62 sites with at least 15 visits (Fig. 1; Table 1). Additional details on the survey and monitoring, as well as additional maps can be found on the PSSS website, www.seabirdsurvey.org.

Figure 1 Minutes of sampling effort recorded (across all observer pairs and months in the Puget Sound Seabird Survey that are included in our analysis), 2007–2014.

Table 1 Name, latitude, and longitude of the 62 sites included in our analysis.

Site	Lat (°N)	Lon (°W)	Site	Lat (°N)	Lon (°W)	
1. 60th St Viewpoint	47.6723	122.4062	32. Mee Kwa Mooks	47.5637	122.4070	
2. Alki Beach	47.5784	122.4144	33. Mukilteo State Park	47.9478	122.3071	
3. Boston Harbor	47.1396	122.9029	34. Myrtle Edwards Park	47.6268	122.3775	
4. Brace Point	47.5152	122.3964	35. Narrows Park	47.2671	122.5641	
5. Brown’s Point	47.3058	122.4443	36. Normandy Beach Park	47.4116	122.3401	
6. Burfoot County Park	47.1310	122.9046	37. North Redondo Boardwalk	47.3507	122.3238	
7. Carkeek Park	47.7125	122.3796	38. Olympia waterfront	47.0582	122.9020	
8. Cromwell East	47.2709	122.6110	39, Owens Beach Pt Defiance	47.3128	122.5280	
9. Cromwell West	47.2714	122.6191	40. Penn Cove Pier	48.2228	122.6883	
10. Dash Pt State Park	47.3204	122.4141	41. Penrose State Park	47.2601	122.7450	
11. DeMolay Boys Camp (E)	47.2777	122.6662	42. Pier 57	47.6062	122.3429	
12. DeMolay Boys Camp (W)	47.2775	122.6668	43. Pier 70	47.6149	122.3573	
13. Discovery Park West	47.6674	122.4227	44. Point No Point	47.9122	122.5265	
14. Dumas Bay Park	47.3263	122.3853	45. Pt Wilson	48.1441	122.7538	
15. Duwamish Head	47.5954	122.3876	46. Purdy Spit South	47.3817	122.6348	
16. Edmonds north	47.8114	122.3891	47. Raft Island north	47.3318	122.6700	
17. Edmonds south	47.8033	122.3947	48. Raft Island south	47.3261	122.6675	
18. Elliott Bay Water Taxi Pier	47.5898	122.3800	49. Richmond Beach	47.7636	122.3858	
19. Fox Island Fishing Pier	47.2287	122.5898	50. Ruston Way	47.2948	122.4990	
20. Frye Cove County Park	47.1152	122.9643	51. Saltwater State Park	47.3728	122.3249	
21. Golden Gardens	47.6928	122.4056	52. Seahurst Park	47.4781	122.3638	
22. Howarth State Park	47.9642	122.2407	53. Sinclair Inlet	47.5398	122.6621	
23. Jack Hyde Park	47.2758	122.4622	54. South Redondo Boardwalk	47.3434	122.3328	
24. Kayak Point State Park	48.1373	122.3668	55. The Cove	47.4428	122.3563	
25. Kopachuck	47.3101	122.6874	56. Thea’s Park	47.2620	122.4398	
26. Les Davis Pier	47.2836	122.4813	57. Three Tree Point	47.4522	122.3792	
27. Libbey Beach County Park	48.2322	122.7668	58. Titlow Beach	47.2469	122.5536	
28. Lincoln Park	47.5263	122.3949	59. Tolmie State Park	47.1209	122.7761	
29. Lowman Park	47.5403	122.3974	60. Totten Inlet	47.1540	122.9645	
30. Luhr Beach	47.1008	122.7272	61. West Point north	47.6624	122.4335	
31. Magnolia Bluff	47.6313	122.3954	62. West Point south	47.6610	122.4330	

Species selection

Over the first seven years of the PSSS (the most recent ending in spring 2014), observer teams recorded 75 unique seabird species. While many of these species may be useful as indicators of various ecosystem processes or human impacts, we focused our analysis on 18 species that have previously been identified as useful seabird indicator species in the region (Table 2; Pearson & Hamel, 2013) because of their relative abundance and dependence on the marine waters of the Puget Sound (Gaydos & Pearson, 2011). These species can be aggregated into five distinct groups: alcids, cormorants, grebes, loons, and waterfowl. Some of the species breed locally in Puget Sound, while others are transient in the Sound, breeding elsewhere (Table 2). Similarly, the species represent a range of diets and behaviors (Pearson & Hamel, 2013), from piscivores (alcids, loons, grebes, cormorants) to omnivores (seaducks and other waterfowl).

Table 2 The 18 species included in our analysis of the Puget Sound Seabird Survey. Rows in bold represent species that breed locally (in Puget Sound).

Common name	Scientific name	Group	
Common murre	Uria aalge	Alcids	
Marbled murrelet	Brachyramphus marmoratus	Alcids	
Pigeon guillemot	Cepphus columba	Alcids	
Rhinoceros auklet	Cerorhinca monocerata	Alcids	
Brandt’s cormorant	Phalacrocorax penicillatus	Cormorants	
Pelagic cormorant	Phalacrocorax pelagicus	Cormorants	
Horned grebe	Podiceps auritus	Grebes	
Red-necked grebe	Podiceps grisegena	Grebes	
Western grebe	Aechmophorus occidentalis	Grebes	
Common loon	Gavia immer	Loons	
Pacific loon	Gavia pacifica	Loons	
Red-throated loon	Gavia stellata	Loons	
Brant	Branta bernicla	Waterfowl	
Bufflehead	Bucephala albeola	Waterfowl	
Common goldeneye	Bucephala clangula	Waterfowl	
Harlequin duck	Histrionicus histrionicus	Waterfowl	
Surf scoter	Melanitta perspicillata	Waterfowl	
White-winged scoter	Melanitta deglandi	Waterfowl	

Statistical modeling

For each species, we constructed matrices of presence–absence, dimensioned by the number of unique month-year combinations (t = 49) and sites (n = 62). Sites that were not visited during a given month were treated as NA values. We constructed a spatial occupancy model separately for each species to incorporate spatial patchiness as well as annual and seasonal variation. The model describing the probability of species presence can be represented as zi,j ∼ Ber(ϕi,j), where zi,j represents the unobserved presence–absence (1, 0), and logit(ϕi) = BXi + ETi + ε, where ϕi represents the site-specific occupancy probabilities at time i, Xi represents a matrix of covariates (Intercept, Month, Month2, Year, Y ear2), B represents a vector of estimated coefficients (shared across sites and time periods), E represents a linear offset coefficient for sampling effort (Ti), and ε represents a vector of spatially correlated random effects. We included time spent (Ti in minutes, ranging from 15 to 60) as a measure of effort to account for the higher chance of recording a species present during longer visits. The spatially correlated random effects are assumed to have the distribution ε ∼ MV N(0, Σ). For simplicity, we modeled the covariance matrix Σ as an exponential covariance function, Σi,j = σ2⋅Ii,j + τ⋅exp(−di,j/γ), where I represents an identity matrix, di,j is the Euclidian distance between sites i and j, and the scaling parameters (τ, γ) control how quickly covariance decays as a function of distance (Banerjee, Gelfand & Carlin, 2005; Ward et al., 2012). Our model could be modified to include more complex covariance functions (Cressie & Wikle, 2011) or spatial random effects that also vary temporally (Shelton et al., 2014). Because our model also includes an observation error component, however, we chose to make these spatial deviations temporally constant. The observation model, linking latent unobserved states (zi,j) to data (yi,j) can be written as yi,j|zi,j ∼ Ber(p⋅zi,j) (Royle & Kery, 2007), where p represents the probability of detection when a species is present.

All Markov Chain Monte Carlo (MCMC) estimation was conducted in R and JAGS (Plummer, 2003; R Development Core Team, 2014), using the R2jags package (Su & Yajima, 2014). We ran five parallel MCMC chains for each species, with a burn-in of 100,000 draws and additional sampling of 50,000 MCMC draws. Trace plots were used to visually assess convergence, and the Gelman–Rubin statistic (Gelman & Rubin, 1992) was used to quantify successful convergence. Not surprisingly, the only parameters that did not successfully converge (potential scale reduction factor >1.05) were several latent states (z) at sites that were not visited by observers in certain months. For the purposes of visualizing predicted hotspots of occupancy in Puget Sound, we used our model output to generate predictions (spatial maps, temporal trends) of species occupancy for a standardized 15 min survey. In addition to making these predictions for each of the 18 species included in our analysis, we attempted to identify hotspot areas, or the sites whose estimated occupancy probability was above the upper quartile (75%) across all sites. Finally, we generated specific occupancy probabilities for the five seabird groups: alcids, cormorants, loons, grebes, and waterfowl. For each group, the probability of occupancy for a group (corresponding to any species from that group being present) was calculated as 1−∏i=1sp1−ϕi.

Results

Our species occupancy maps reveal localized hotspots of occurrence in Puget Sound for some alcid and cormorant species (rhinoceros auklet, pelagic cormorant, Brandt’s cormorant (Fig. 2) as well as for loons and some waterfowl species like harlequin ducks (Fig. 3)). The individual species maps show that some species are ubiquitous in all nearshore habitat (horned grebes, goldeneyes, scoters), while others have a much more patchy distribution of occurrence (loons, rhinoceros auklets, pigeon guillemots). These patterns become even more apparent when we examine the sites in the upper quartile (75–100%) of the estimated occupancy probabilities across sites (Figs. 4 and 5). Some maps of very rare or very common species may not be informative, but areas of high bird density become more apparent when our estimated occupancy probabilities are calculated by group (Fig. 6). For example, each loon species in the survey is relatively rare (Fig. 3), but the aggregated spatial distribution of all loons shows several patches of high and low occurrence, with the highest density of occurrence in the central-south Puget Sound (Fig. 6).

Figure 2 Estimated probability of occurrence for the 62 sites included in our analysis. Presented estimates are for alcids, cormorants, and grebes in December 2013. The color scale used to represent occurrence probabilities ranges from 0 (a species is not present) to 1 (occurrence is 100%).

The 18 species included in our analysis showed a range of seasonal variation, with waterfowl species (bufflehead, common goldeneye, surf scoter) and grebes varying the most and peaking in December–January (Fig. 7). Although most of the 18 species had relatively small variation over the 7-month survey period, several species exhibited monotonic increases (pelagic cormorant, pigeon guillemot). Of the 18 species, the probabilities of trends in occurrence being positive over the 7-year survey were greater than 80% for 14 species (Fig. 8). Western grebes, white-winged scoters, and brants showed relatively strong negative trends in occurrence (probabilities of negative trends >99%, 84%, 79%, respectively).

The 18 species in our analysis represent a gradient of occurrence probabilities and trends over space. Several species from each group were relatively rare in central and south Puget Sound; the rarest species included two of the alcids (common murre, marbled murrelet), western grebes, all three loon species, and three of the waterfowl species (brant, harlequin duck, white-winged scoter; Figs. 2, 3 and 8). In contrast, horned grebes and three different waterfowl species (bufflehead, common goldeneye, surf scoter) were the most widely occurring (Fig. 8).

Figure 3 Estimated probability of occurrence for the 62 sites included in our analysis. Presented estimates are for loons and waterfowl in December 2013. The color scale used to represent occurrence probabilities ranges from 0 (a species is not present) to 1 (occurrence is 100%).

Figure 4 Estimated hotspots of occurrence for the 62 sites included in our analysis, defined as probabilities in the upper quartile (75–100%) across sites (Fig. 2). Presented estimates are for alcids, cormorants, and grebes in December 2013. The color scale used to represent sites in the upper quartile is red (>75%) or white (<75%).

Figure 5 Estimated hotspots of occurrence for the 62 sites included in our analysis, defined as probabilities in the upper quartile (75–100%) across sites (Fig. 2). Presented estimates are for loons and waterfowl in December 2013. The color scale used to represent sites in the upper quartile is red (>75%) or white (<75%).

Figure 6 Aggregated probabilities of occurrence for each of the five groups in our analysis, as well as for all 18 species. For groups, these represent the probability of seeing any bird that is a member of that group; for all species, these represent the probability of seeing at least 1 bird (of the 18 species in our analysis). Estimates are shown for December 2013. The color scale used to represent occurrence probabilities ranges from 0 (not present) to 1 (occurrence is 100%).

Figure 7 Estimated median probabilities of occurrence by month. Estimates are shown for the most recent year (October 2013–April 2014). Estimates for November, January, and March are not shown.

Figure 8 Estimated probability of occurrence in the 2013–2014 seabird survey (with 25%, 50%, 75% intervals), percent change in the probability of occurrence from 2007 to 2013 (25%, 50%, 75% intervals), and the probability of the annual rate of change from 2007 to 2013 has been positive. All data (2007–2013) are used to estimate intra- and inter-annual trends.

Discussion

Analyses that incorporate both spatial and temporal variation are becoming increasingly common in ecology. These types of analyses are widely applicable to virtually any type of observed data, from presence–absence to continuous observation measurements (Johnson et al., 2013; Shelton et al., 2014). Incorporating spatially structured random effects introduces a layer of statistical complexity that is warranted in many cases because predicted density estimates (both in space and time) are more precise (Thorson et al., in press).

Spatially-structured citizen-science datasets have been used at a large spatial scale, particularly in quantifying shifts in phenology linked to climate. One of the most frequently documented changes by citizen-science efforts has been shifts in breeding seasons (Hitch & Leberg, 2007; Hurlbert & Liang, 2012; Mayer, 2010). Spatially-structured statistical models have been fit to these types of datasets to improve estimates of trends (Hurlbert & Liang, 2012; Thorson et al., 2014), but few analyses have applied spatiotemporal models to data from citizen-science efforts to identify hotspots or areas of conservation concern at a fine spatial scale. Citizen-science programs, such as the Puget Sound Seabird Survey data analyzed here, offer a unique opportunity because both the temporal and spatial scales of data collection are much finer than national (Breeding Bird Survey) or regional (WA Department of Fish and Wildlife) efforts. If volunteer-driven science can result in relative indices of occurrence or abundance, it provides an extremely cost-effective approach for identifying local areas of risk (Hass, Hyman & Semmens, 2012) or potential hotspots of diversity that may be useful in conservation planning (e.g., establishing reserves) or permitting activities.

Using citizen-science data—either to complement existing datasets or to fill in data gaps when other surveys are absent—is particularly important for areas or habitats at risk. The PSSS may be a good model for adopting similar citizen-science efforts, either in other regions or for other applications that may also be used to study food webs—examples include monitoring water quality and the spread of invasive species (Silvertown, 2009). In addition to the historic decline of many seabird species (Bower, 2009), there are a number of other human impacts that have caused shifts or reorganization in the prey base (Blight et al., 2014) or competitors of seabirds (Harvey, Williams & Levin, 2012). These impacts could include effects of overfishing or bycatch (and associated impacts of derelict fishing gear; Good et al., 2009), climate change, toxins (Good et al., 2014), habitat loss (Raphael et al., in press), altered freshwater flow regimes, and the recovery of many top predators to historic levels (pinnipeds, harbor porpoise, bald eagles).

Although many seabird species in the Puget Sound region are thought to be depleted relative to abundances in the 1960s–1970s (Bower, 2009; Vilchis et al., 2014), our results present a more optimistic picture for a number of species over the last decade. Of the 18 species included in our analysis, we found strong support for 14 having increasing probabilities of occurrence, and these results are in agreement with recent studies in the region (for example, nesting surveys suggest Rhinoceros auklets are also increasing; Pearson et al., 2013). Many of the species that are occurring more frequently are those that breed in the region (Table 2). In the list of indicator species compiled by Pearson & Hamel (2013), some of these species (scoters, murrelets) were declining significantly when considering trends based on total abundance, so it is possible that species in decline have a less aggregated spatial distribution, resulting in their probability of detection increasing. Another possibility is that the PSS survey measures occurrence close to land, while trends from other surveys may represent slightly different habitats. Of the species not increasing, one species provided weak support for declining occurrence (white-winged scoter), and three species provided strong support for continued declines in occurrence (brant, western grebe, red-necked grebe). These three species in decline are also concerning because they are already rarely seen species in the PSSS data (Fig. 8).

There is no obvious mechanism for why the three declining species in our analysis exhibit a declining trend in occupancy, but some of these declines may be occurring at breeding colonies (not in Puget Sound) or resulting from shifts in prey abundance in the Puget Sound region. Some recent evidence suggests that there have been long-term changes in the base of the food web of the Salish Sea (Blight et al., 2014), and over-wintering seabird species that rely on forage fish are declining (Vilchis et al., 2014). Another mechanism that may also be related to shifts in the spatial distribution of prey is the large-scale shifts in seabird species’ ranges. For example, Wilson et al. (2013) used citizen-science data to show that western grebes appear to have shifted out of Puget Sound region to the southern end of the California Current. Our estimated declines in occupancy over the last seven years are largely in agreement with a continued decline in the occurrence of western grebes in the region. Like western grebes, brants and white-winged scoters over-winter in Puget Sound but breed elsewhere, and thus may be affected by threats in other ecosystems. Though the exact mechanisms responsible for these trends are not known, our trend estimates may be useful in prioritizing monitoring efforts or refining existing marine bird or ecosystem indicators in the region (Kershner et al., 2011; Pearson & Hamel, 2013).

Although the focus of our volunteer-driven surveys in the Puget Sound region are focused on identifying spatial hotspots and improving estimates of annual trends, citizen-science efforts like the PSSS may provide additional valuable baseline monitoring. The 7-year dataset analyzed here provides both a baseline for seabird monitoring in 2014, and also allows us to do a retrospective analysis of trends over this time period. For example, in the event of an oil spill in the region, PSSS data could provide 7 + years of pre-spill information on seabird distribution and abundance for comparison. Having a 7-year period as a baseline instead of just a single year is useful in that the year-to-year variability can be quantified. Such citizen-science efforts may also be scalable to different types of data collection that also involve spatially structured threats to marine ecosystems such as harmful algal blooms, ocean acidification, and fisheries activities.

The Puget Sound Seabird Survey was conceived, developed and implemented by the Seattle Audubon Society and continues to be a major focus of their science efforts. We thank the 250 + observers who braved wind and rain to collect bird data; this work would not be possible without their dedication. Thanks to J Smith, C Jordan, T Good, and two anonymous reviewers for providing helpful reviews of earlier drafts of this manuscript.

Additional Information and Declarations

Competing Interests

Author Contributions

Eric J. Ward is an employee of NOAA, Kristin N. Marshall is a post-doc at NOAA, Scott F. Pearson is an employee of Washington Department of Fish and Wildlife, Nathalie Hamel is an employee of Puget Sound Partnership, Adam Sedgley and Toby Ross performed this work as employees of Seattle Audubon.

Eric J. Ward and Kristin N. Marshall conceived and designed the experiments, analyzed the data, contributed reagents/materials/analysis tools, wrote the paper, prepared figures and/or tables, reviewed drafts of the paper.

Toby Ross, Adam Sedgley, Todd Hass, Scott F. Pearson, Gerald Joyce, Nathalie J. Hamel, Peter J. Hodum and Rob Faucett conceived and designed the experiments, contributed reagents/materials/analysis tools, wrote the paper, prepared figures and/or tables, reviewed drafts of the paper.

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
