# Peer review of "Using citizen-science data to identify local hotspots of seabird occurrence"

_PeerJ, doi:10.7717/peerj.704_

## Round 0.1 · original submission · Minor Revisions

The Paper needs minor Changes before its publication. The Introduction and Discussion should be re-worked before the paper is recommended for publication. As is, it does not connect well to what the authors proposed to achieve.Please also follow suggestions made by both reviewers that will make your Ms. better, including new citations suggested, changes in Tables and Figures and clarifying the text in those points they have remarked.

Reviewer 1 ·

Basic reporting

Overall the article is clearly written with good English. It s a good study that is needed for science-based conservation of Puget Sound.

ABSTRACT
The abstract summarizes well the Introduction, Results and Discussion by highlighting key points from each of these sections. There is an incomplete sentence in line 3; the word ‘because’ is followed by a period and no text. I think the Methods highlights could be a bit better in the abstract. As written in only describes the Puget Sound Seabird Survey as a data set, while in the rest of the manuscript plenty of text is used to describe this survey. I would suggest a sentence of describing what the Puget Sound Seabird Survey is and its time span and area coverage. You could save word count by removing ’62 sites in the current analysis’ in sentence 9. It is too detailed for an abstract. Stating the two grebe species that are declining in occurrence would make the abstract read better.

INTRODUCTION
I found the introduction to be a bit long with the information not leading directly to the purpose of the study. All of the information is there, I just suggest a bit of trimming and re-arranging and of the text. The statement of purpose in the abstract does match well with what is written in the introduction. As written, arguing for validity and importance of the PSSS overshadows the argument for conducting the study. To help trim down the introduction, some of the details of the PSSS could fit better in the methods section.

Line 96 — You cite Anderson 2009 – Marine Ornithology to describe the WDF&W aerial surveys. A better description/resource and thus citation of these surveys is Nysenwander 2005. It is what Anderson 2009 and Vilchis 2014 also cite.

Line 105—The authors describe the study’s purpose of establishing a baseline monitoring of local seabird occurrence and abundance in winter months. WDF&W aerial surveys however, began in 1994 and are ongoing, so baseline monitoring already exists in Puget Sound. A caveat of the WDF&W aerial surveys is that they take a synoptic picture of Puget Sound while surveying Puget Sound in six weeks or so. PSSS surveys while smaller in spatial scale have the advantage of sampling birds repeatedly during October to April at specific sites in Puget Sound. Nevertheless, the devil is in the details…and while Oct-Apr obviously includes winter, it also includes two months of fall and one month of spring. I would suggest rewording baseline monitoring in winter months something more appropriate, like longitudinal sampling of seabird abundance from late fall to early spring on a yearly basis from 2007 to 2014, as you do in the methods.

METHODS
The way the methods are written it sounds like the authors specifically designed the PSSS for this study, whereas you are using PSSS data for this study. If the former is correct then this should be mare clear in the Introduction. I would suggest having a paragraph with a PSSS subheading that describes the survey and perhaps referencing the program to its website (http://www.seattleaudubon.org/sas/WhatWeDo/Science/CitizenScience/PugetSoundSeabirdSurvey.aspx) or some published document that thoroughly describes the monitoring program.

Line 128—I assume this means 300 meters from shore, but this should more clear.

Line 136—Again, this sounds like the emphasis is completely on why the PSSS exists and its importance. The authors objectives were to evaluate relative hotspots of seabird occurrence in Puget Sound using data from the PSSS collected from 2007 to 2014. Stating this in the methods in this manner I think would make the manuscript more clear. As written the reader is left thinking where is the rest of the time series.

Line 144—I think it would be more reasonable to classify cormorants as piscivorous birds with diverse fish diets versus omnivorous birds. See Ainley and Kelly 1981 Condor ad Ainley and Sanger 1979 in Conservation of marine birds of northern North America. If I am wrong, then please site the reference for this statement.

Line 146—If space permits, a bar graph (or something as such) describing the yearly frequency of sampling at each the of the 62 survey sites would be beneficial to show that effort was equally distributed throughout all 62 sites.

Line 199—‘Monotonic increases throughout winter’. Is this from January to March or to October to April? You can see why this is confusing. The authors continue on describing the 7-month survey, so I am assuming they are referring to the latter time period. This does not correspond to winter alone. Clearing this up as mentioned above would benefit the manuscript.

RESULTS
All titles and legends accurately describe content the content and the text complements well the data without simply repeating it. The legends of figures 1-3 should have a label saying ‘Probability’. The results shown in Figures 1-3 depict only 58 of the 62 survey sites described in the Methods section (Line 147 and Table 1). One is left wondering why Libbey Beach County Park, Penn Cove Pier, Pt Wilson, Kayak Point and State Park (the four northernmost sites) were left out of the maps and results. I am sure there is a good reason, but this should be made clear in the results, or these sites four sites should be included in the maps. A map showing they study area along with the entire 62 PSSS survey sites analyzed for the study would complement well table two. If space permits, both should be included in the final manuscript. If space is limiting, then I would recommend the map is used instead. For readers not familiar with the region (as most of PeerJ’s audience probably are), one could assume that Hood Canal was also sampled when in was not. Only sites in south and central Puget Sound were sampled, along with a few northern ones near Admiralty Inlet and Whidbey Island. Clearing this up from the get go in the methods under a ‘Study Area’ subheading would make the paper much more clear.

DISCUSSION
The first three paragraphs of the discussion do not rise logically from the data. These three paragraphs read more like they belong in the Introduction or Methods section, or perhaps in condense fashion at the end of the Discussion, but not at the beginning. These paragraphs are well written, but they make the case of using spatial temporal analyses and citizen science data, which is great, but an argument better suited for the end of the discussion, or the conclusion and the end of the paper (like the authors have already). Beginning the Discussion by interpreting the results would make the manuscript read much better. As a result, I found the Discussion section lacking in interpreting the very interesting patterns the authors found. For example, why do the authors think their results show 14 species becoming more common in Puget Sound, when Bower 2009, Anderson 2009 and Vilchis 2014 report otherwise? Why are bird species that breed locally (within Puget Sound) are not declining? Using the Discussion to more elaborately interpret the study's results and perhaps develop some hypotheses explaining why authors found differences would make the Discussion more robust and greatly benefit the paper.

Line 253—This line of reasoning does not make sense. Please spend a few sentences elaborating on this to explain and re-enforce the statement. Otherwise I would suggest deleting it. Where else have we seen this phenomenon occur?

Line 261—Here you state ‘recent evidence suggests that there have been changes in forage fish in the region’. It looks like the authors are suggesting temporal changes in forage fish abundance/availability in the region. If this is the case, then this line of reasoning is flawed because Rice et al 2012 trawled Puget Sound in May-August of 2003. Four years before the time span of this study and for only one year and during summer. Rice et al 2012 did find spatial difference in the composition of their trawls, with more fish in the trawls in the northern part of Puget Sound. If this is what the authors are referring to, then it should made clear in the text.

Line 261— A space between ‘the’ and ‘spill’ is needed.

Experimental design

Great design with a novel approach of interpreting data generated by the Puget Sound Seabird Survey.

Validity of the findings

The data of this study seems robust and statistically sound, however the conclusions could be stated more clearly by connecting the results to the original question sated in the introduction of the manuscript. As I say above, re-working the Discussion as to interpret the study's results, discuss why these are different from previous studies in the region and connect these the study's objectives would significantly increase the validity of this study.

·

Basic reporting

Abstract (no line numbers provided)

General: why are you assessing spatial patchiness and interannual trends? The first two sentences are general and do not apply to the study area.
Scale: Is scale an important message for this study? If so, why then is this relevant to the seabird populations being examined? Are you considering the Puget Sound as “large spatial scale” or is it a “fine scale”? I’m not clear on what you think the scale is of this study. What about other studies that have been done for the entire Puget Sound, such as the recent Vilchis et al (2014) or Hamel et al (2009), what scale are they?
Anthropogenic activity – simply say human activity
Indicators – see comment above.
Fourth line – sentence ends with …. “because .” This sentence isn’t finished.

Introduction

General: Interesting background information. I felt the introduction seemed to jump a bit from topic to topic. I think a bit of smoothing out and using topic sentences would help. For example, I wasn’t sure why I should care about the lack of spatial-temporal models in ecology and whether this paper was then going to be a methods paper (e.g., a paper suitable for Ecological Applications) or if it was an applied paper to answer a particular management question.
Line 42. I think resource managers would be included here, based on the next sentence.
Lines 46-51. The leadoff sentence on line 45 says this has implications for (or, is of interest to) science and management. The next three sentences lead off with “conservation perspective”, “theoretical ecology perspective” and the last one is not qualified as to whether this is management or science. I think making the connections to four thematic areas, not two, could strengthen this paragraph: conservation, research, management, and policy.
Line 52. The word “trend” implies time, so no need to say “… through time”. There are several places in the manuscript where this applies.
Line 52 thru 65. I think that it might be possible to combine these two paragraphs and specifically talk about this study determining “probabilities of occurrence”, which isn’t actually mentioned in the introduction. I don’t really think the focus of this paper is about the model per se so I don’t think it’s all that relevant to have an entire paragraph on this; the model is a tool for you to be able to explore the data relative to your hotspot question, right?
Line 55. For a marine paper, seems a little odd to have an example as wildfire.
Line 57. Marxan is another example of a computation or algorithm that explores spatial and temporal trends and provides a spatial output.
Line 63. As a reader, I was wondering how you defined observation error and why does it matter? This ties to a statement on Line 109, about “ongoing validation to quantify bias” and why this makes the PSSS an excellent program.
Line 65. It might help here to conclude this paragraph with a statement of what or how this study is intending to contribute to, or advance, this aspect of quantitative ecology.
Line 66. What is the segue from models to long-term monitoring? This is an example of a jump in the introduction.
Line 94. How long has the Anderson study been going?
Line 100. See also: Anderson et al. 2009. Using predator distributions, diet, and condition to evaluate seasonal foraging sites: sea ducks and herring spawn. Marine Ecology Progress Series. 386: 287-302
Line 101. This sentence seems to dilute or contradict the discussion in that the cause of the decline of the piscivorous birds points to decline in forage fish, as per the Vilchis et al (2014) study.
Line 108. “Represents one of the more rigorous citizen science efforts.” Is there a geographical qualifier for this statement…more than other studies in WA state, United States, North America, the world?

Tables & Figures

Line 291. Marbled Murrelet mostly breed in the coastal old-growth forests of WA state, including surrounding Puget Sound but also along the outer coast. I think that some of their nest sites are probably as close as the murre nest location on Tatoosh. The term “local” is interesting because the loons breed on lakes in WA state, as well as the buffleheads and goldeneye, which is local relative to those species that breed in the arctic.
Line 297. I like the plots. They are easy to read. Only comment is that there were no survey locations in Hood Canal so you might think about colouring those waters differently, for within the PSSS area vs. outside?
Line 297. The red scale for probability should probably be labeled, or mentioned in the caption, that it goes from 0-1, with 1 indicating 100% probability.
Line 306. Figure 3. What are the aggregated probabilities showing us? The aggregation appears to make everywhere of high probability so there are no defining areas of importance. How does this plot help your story, esp. the one for “All species”?
Line 307. When you say “all species”, is that all 75 or just the 18 indicator species?
Line 318. I think it would help to label your plot with (a) (b) and (c) so that you can specifically refer to one of the sub-plots in your Results.

Experimental design

Methods

Line 120. A short description of Puget Sound and map would be helpful for readers that are not familiar with this geography. On this map, it would help to define what you mean by “local” to support Table 2 for the breeders (Line 291). The study site description might include that Puget Sound and Salish Sea are important overwintering habitat for 70+ bird species; there number of listed bird species, human population trends and shoreline development, forage fish losses, possibly endangered fish species (rockfish).
Line 123. Seabirds - Consider using the term marine birds instead of seabirds. Mergansers, buffleheads, and seaducks are not “seabirds” and marine birds is a nice general term.
Line 124. Might be helpful here to explicitly say that the survey was not done year round because these birds are not generally present in the Puget Sound in the summer months because they are breeding elsewhere. It’s implied by the sentence and the caption in Table 2 but the point that the PSSS targets birds in the non-breeding birds, and only some of them breed in the area could be lost.
Line 182. Similar to the equations above, do the equation components for the probability need to be defined?
Line 182. Is the probability of occurrence relative to 2013/2014? Is that why in Figure 5a, the probability of occurrence is for 2013/2014 but then there are the inter-annual trends presented for the 6 years of the study in 5b and 5c?

Validity of the findings

Results

Line 149 to 151 – Is the model only presence-absence? I don’t understand how it takes into account abundance. For example, a species with a high population has a higher probability of being encountered than a species with low population. Does “patchy distribution” mean spatially patchy or numerically patchy, or both? You could have high patchiness and high abundance, like the Western Grebes off Carkeek Park.
Line 187. Referring to Figure 1, the caption says December 2013 however I was expecting to see multi-year results because the methods note that the model incorporated annual and seasonal variation (Line 150). Do these plots represent just one year? If so, then I am not clear on the methods. Perhaps, in Lines 176-178, something could be added here to clarify the temporal aspect of the spatial plots?
Line 202. I don’t see the negative trends in Figure 5. The x-axis contains only positive values.
Line 204-210. The results refer to Figure 5 however the Figure is not spatial thus the statement that several groups were reare in central and south Puget Sound cannot be verified from this figure.

Discussion

Line 247. The Vilchis et al (2014) study used data from a longer time series I think, so might want to be careful about the optimism. Also, the PSSS does not cover all waters of Puget Sound, just the first 300 m of shoreline.
Line 251. Is this reference for Pearson and Hamel (2013), not Pearson et al (2013)?
Line 257. Are some or all of these three species mostly beyond the 300 m semi-circle of PSSS? For example, the Western Grebes are nearly always offshore.
Line 260. For these three species, the declines could be occuring on the breeding colonies?
Line 272. This conclusion paragraph could be stronger with respect to the relevance of the study, and also tie back to the indicators mentioned in the first sentence.
Line 273. Why are highest probabilities generally on the eastern shore of Puget Sound or southern shore of the peninsula (Fig 3)? This is not explained in the discussion.

Additional comments

Really great to see the results of the PSSS being published. I think this is a valuable project for Puget Sound, and brings good information to bear on the overwintering location of marine birds in the Sound. I think this type of research is also relevant for the marine bird monitoring programs in Canada, and joint Canada-US work in the Salish Sea. I think that this paper strengthens the case for citizen science, showing that these programs can be designed to produce meaningful results. The six year time series of the study is excellent and could be emphasised more strongly than it is now in the paper.

Recommend, with minor revisions. No major revisions to methodology or analyses required. Suggestions on revising text for clarity, presentation and flow, as well as considering alternative hypotheses for the results.

General comments on the manuscript for the authors to consider:

Indicators – The first sentence of the abstract is about using seabird as indicators, but this doesn’t really seem to be the theme of the manuscript. There are only a few references to indicators, Line 116 and 140, and so I’m not really sure how this work ties into the management of these indicator species. The theme of this paper seems to be about identifying areas of repeated high abundance or occurrence (using probability theory). If the authors want a connection to indicators, then it would be good to bring this into the introduction and discussion. Couple of questions that might help to guide this: How are seabirds being used as indicators in the PSSS study, and what are they indicating in the marine environment? Are these seabirds good indicators of local conditions or are they indicators of something happening over a much larger area? Is there a desire to have this research connected to the Puget Sound Partnerships’ Vital Signs? If so, what are those connections and how does the PSSS strengthen the Vital Sign work to track the progress of restoring Puget Sound?

Hotspots –I think this term has kind of gone out of favour because you don’t see it that much in the conservation literature now. In conservation, terms like “high priority areas” or “high value areas” to reflect that the data are being used to inform a management or conservation question rather then a theoretical science question.

Habitat – In addition to prey availability, another hypothesis with respect to the results is site fidelity, or philopatry, relative to the nearshore habitat characteristics. This is mentioned in the Introduction (Line 52-53). There are several variables that might explain the probabilities of occurrence including: amount of exposure to wind and waves, presence of predators (e.g., eagles), substrate and vegetation, and disturbance from humans (e.g., dogs). Most of these data were collected by the PSSS volunteers and I think these variables might also explain some of the results. The at-sea variables in the discussion are valid but since the surveys occur close to shore (300m), then physical habitat may be more important because the birds weren’t foraging, they were resting or socialising. Thus, the distribution of birds may reflect those areas of the Sound with limited shoreline development, that is, relatively unaffected areas with good places to rest (e.g., shallow bays). Many of the survey sites were public parks or beaches however there was a wide variety in the levels of shoreline development. Of course, there is the problem that the sites were not randomly chosen so what can be said about habitat will be speculative.

Conservation objectives – I think that with respect to managing the seabird populations in Puget Sound you are talking about conservation objectives for management, or natural resource management objectives, these aren’t conservation objectives for conservation organisations (like The Nature Conservancy or National Audubon Society). I think it would be important to tie these statements to a specific policy that you are hoping to inform, if it exists. For example, does WDFW or DOE have a management objective for the seaduck populations (e.g., scoters) that they do not fall below some targeted population level?

Terminology and Focus – A mix of terminology is used in this paper and it might confuse some readers both in terms of scientific jargon and focus – hotspots, indicators, temporal and spatial trends, probabilities of occurrence, baseline data, citizen science, ecological models. I might suggest that you think about how you can simplify and sharpen the focus. For example: Is this paper about creating a baseline data for 2014 so that future changes can be detected and monitored? Is this paper a retrospective analysis to determine past trends and consider management and policy implications given the current population levels? Is this paper about developing a particular mathematical model that advances quantiative ecology and solves some of the computation problems (e.g., incorporating underlying vulnerability to avoid poor estimateion of trends? Line 56) I think it’s okay if there are multiple objectives for the paper, but it might help to spell these out clearly.

---

## Round 0.2 · accepted · Accept

The authors have included the suggested bibliography, have changed tables and figures and rewritten the introduction and discussion main changes suggested by both reviewers